# Practices of Rapid Sequence Induction for Prevention of Aspiration—An International Declarative Survey

**DOI:** 10.3390/jcm14072177

**Published:** 2025-03-22

**Authors:** Imen Ben-Naoui, Vincent Compère, Thomas Clavier, Emmanuel Besnier

**Affiliations:** 1Department of Anesthesiology and Critical Care, Centre Hospitalier Universitaire de Rouen, F-76000 Rouen, France; imen.ben-naoui@chu-rouen.fr (I.B.-N.); vincent.compere@chu-rouen.fr (V.C.); thomas.clavier@chu-rouen.fr (T.C.); 2Univ Rouen Normandie, Inserm U1096, F-76000 Rouen, France

**Keywords:** anesthesia, pneumonia, aspiration, rapid sequence induction and intubation, surveys and questionnaires

## Abstract

**Background/Objectives**: Rapid sequence induction (RSI) for the prevention of aspiration is a frequent clinical situation during anesthesia. The lack of international guidelines on this topic may lead to differences in practices. The aim of this survey is to identify the clinical practices in RSI among practitioners at an international level. **Methods**: International declarative survey across the ESAIC network. **Results**: A total of 491 respondents in 61 countries, 74% of them were seniors and 42% with over 20 years of experience. Most of the practitioners (87%) performed preoxygenation under a high flow of oxygen (>10 L/min) with no PEEP and no pressure support and 69% use opioids in most cases of RSI. The Sellick maneuver was used by 42% of respondents. RSI was used in most situations at high risk of aspiration (bowel obstruction, trauma within 6 h after the last meal, caesarian section). RSI was used in 53% of cases of appendicectomy in the absence of vomiting. Conversely, 29% did not use RSI in cases of symptomatic esophageal reflux. A total of 11% encountered at least one episode of grade IV anaphylaxis to succinylcholine or rocuronium and 24% aspiration pneumonia. **Conclusions**: Our results support the need for international guidelines on RSI to limit differences between practitioners and countries.

## 1. Introduction

During the last few decades, many factors have led to a remarkable growth in anesthesia practice, following the evolution of surgical and non-surgical procedures. Meanwhile, perioperative outcomes have been steadily improved over the last few decades, with an estimated incidence of all adverse events inferior to 2% of procedures, and a very low incidence of death, highlighting the current safety of anesthesia [1,2]. Despite these improvements, aspiration remains a devastating complication, accounting for 5% of anesthesia practice in the United States nowadays, with a mortality of more than 50% of aspiration cases [3]. Thus, aspiration may be responsible for up to 9% of anesthesia-related deaths [4].

Rapid sequence induction (RSI) was first described by Sept and Safar in 1970 and aimed to minimize the risk of aspiration. The authors used the combination of thiopental and succinylcholine, allowing a very short period before intubation, and therefore avoiding mask ventilation [5]. They also apply a cricoid pressure to increase esophageal pressure, as described by Sellick in 1961 [6]. Nevertheless, the absence of clinical guidelines for RSI for the prevention of pulmonary aspiration may result in differences in clinical practices.

Guidelines for RSI in the particular setting of critical care were recently published and studies and surveys have raised the issue of practice heterogeneity in RSI [7,8,9].

In order to highlight these differences, we conducted an international declarative survey among anesthetists to analyze practices according to the different situations, professionals, generations and countries.

## 2. Materials and Methods

### 2.1. Ethics

Ethics approval was not required for this study because it did not involve patients or animals and was limited to a questionnaire. According to French law, the study follows the guidelines for data management of the MR-004 methodology of the French National Institute for Informatics and Freedom (Comission Nationale de l’Information et des Libertés—CNIL). The survey was anonymous and physicians were informed of the objectives and participated of their own free will. No written consent was required.

### 2.2. Study Design

We conducted a cross-sectional, international and declarative survey during a one-month period, from 15 April 2022 until 15 May 2022. The survey was sent electronically to the members of the European Society of Anaesthesiology and Intensive Care (ESAIC) via the online software surveymonkey^®^ (https://fr.surveymonkey.com), accessed on 15 April 2022. It was developed according to available guidelines [10]. Briefly, the survey was designed by three senior anesthetists and then tested by five experienced anesthetists before dissemination. Moreover, the survey was validated by two experts from the ESAIC before dissemination. The survey was anonymous and responders were able to skip some questions at their discretion to facilitate completion, which may explain some missing data.

### 2.3. Survey Design

The questionnaire was anonymous. It consisted of 34 questions divided into three parts:-The first part included demographic data about the anesthetist’s country, professional position, field of practice and personal experience.-The second part explored RSI practices, with a sub-section consisting of clinical vignettes where respondents were asked about their attitude in cases where RSI may be questionable. This intended to explore the practices in situations where guidelines may be taken as default.-The third part included questions about critical incidents related to RSI, and eventual legal litigations following these incidents.

The full survey (and its results) is available in Appendix A.

Data are presented as absolute numbers or percentages and medians with first and third quartiles.

### 2.4. Outcomes

The main outcome was the proportion of respondents using an adequate definition of RSI (the use of at least a neuromuscular blocker and a hypnotic drug, with or without opioids). The secondary outcomes were the other practices surrounding RSI (Sellick’s maneuver, stylet, gastric tube, gastric buffering, etc.) and the specific practices for particular clinical vignettes.

### 2.5. Statistical Analysis

Data are presented as absolute numbers and percentages or median with first and third quartiles, as the distribution of data was not normal (as tested using a Shapiro–Wilk test). An analysis of responses by country was also carried out if the number of respondents was greater than 10 per country. However, due to the large number of countries, an imbalance in the number of respondents per country, and the sometimes small numbers involved, no statistical comparison was made.

Missing data were not imputed because of the declarative nature of the survey.

## 3. Results

The survey was sent to 8903 e-mail addresses and 8901 had successfully received the mail after 2 bounce backs, for a total of 3161 (35.5%) practitioners who opened the campaign, but only 588 (6.6%) clicked on the link. A total of 494 anesthetists answered the survey with a response rate of 15, 6% (494/3161). Because the respondents had the ability to skip questions, and because some questions were conditional, the lowest number of answers per question was 208, for a total of 64% completion of questionnaires.

The complete results are available in Appendix A.

### 3.1. Demographics

Respondents were from 61 countries all over the world, mainly from Europe (notably Germany (47/460, 10%), Switzerland (38/460, 8%), Spain (27/460, 7%), Portugal (27/460, 7%), France (26/460, 6%), Belgium (24/460, 5%) and Austria (22/460, 5%)), but also from other continents (for details, see Appendix A).

Most of the respondents were senior anesthetists (364/462, 79%), followed by residents (63/462, 14%) and junior anesthetists (32/462; 7%), with a mean age of 44 (35–55) years old. Up to 42% (194/463) of them presented with more than 20 years of experience.

Anesthesia was the main field of practice for the majority (279/463, 60%) and 36% (168/463) of practitioners had a mixed activity between intensive care and anesthesia. Only 3% exclusively practiced intensive care (13/463). Half of the respondents practiced in university hospitals (240/463, 52%) and 33% (155/463) in general hospitals.

The respondents declared performing 15 [10,11,12,13,14,15,16,17,18,19,20,21,22,23,24,25] anesthesia per week, essentially for general surgeries (abdominal, urology and gynecology; 404/463, 87%), orthopedic surgeries (316/463, 68%), ear, nose and throat surgeries (252/463, 54%), vascular surgeries (211/463, 46%) and thoracic surgeries (129/463, 28%). Other surgeries were practiced by fewer than 20% of respondents. Among respondents, 46% (212/463) declared managing urgent abdominal surgeries more than once a week, 35% (160/463) trauma surgeries more than once a week and 36% (166/462) obstetrics more than once a week.

### 3.2. RSI Practice

Most respondents defined RSI as an association of a short-acting hypnotic, a short-acting neuromuscular blocker with (69%) or without (49%) opioids (Table 1). Only 2% declared not using a neuromuscular blocker. Most of them ask for preventive assistance from another anesthesia provider (physician or nurse) and use a stylet inside the tracheal tube to facilitate intubation. Fewer than half of the respondents declared performing a Sellick’s maneuver. Preoxygenation was performed using a high flow of oxygen (>10 L/min) in the majority of responses (87%), and some respondents declared also using non-invasive ventilation or a high-flow nasal cannula.

Only 20% of respondents declared performing gastric echography in some situations. The main reason for not using this technique is a lack of knowledge or training.

### 3.3. RSI in Various Clinical Settings

In this section, we explored the declared attitude of anesthetists in clinical situations where the use of RSI may be considered. The complete results are presented in Table 2. We explored both the risk of aspiration because of the absence of gastric vacuity or because of a gastroesophageal reflux.

The results concerning situations at risk of the absence of gastric vacuity are heterogenous, depending on the clinical situation. If most respondents declared using RSI in cases of acute bowel obstruction, cesarean section or trauma with fasting < 6 h (respectively, 97%, 89% and 81%), only 53% of respondents declared performing a standard induction (no RSI) in cases of appendectomy with fasting > 6 h and no vomiting. Similarly, in cases of a leg fracture with fasting > 6 h, only 41% used RSI despite a trauma occurring within 2 h after the last meal. The use of gastric suction before anesthesia depended on the situation, ranging from 1% for a trauma with fasting > 6 h to 83% for acute bowel obstruction. Gastric buffering use ranged from 15% (appendectomy) to 66% (planned cesarean section). Prokinetic use was limited (10–31% of cases).

The results for situations at risk of esophageal reflux varied according to the symptomatology of patients. Indeed, 70% of respondents used RSI and 54% gastric buffering in cases of symptomatic reflux. Conversely, only 26% used RSI and 22% gastric buffering in cases of asymptomatic reflux (under pharmacological treatment).

### 3.4. Aspiration, RSI Side Effects and Legal Issues

The last part of the survey explored the complications related to aspiration and/or RSI side effects, and the potential for legal issues. The results are presented in Table 3. Briefly, 29% of respondents declared being implicated in a grade 3 anaphylaxis due to a short-acting drug for RSI, and 11% in a grade 4 anaphylaxis. Conversely, 76% declared being implicated in at least a case of aspiration. The respondents more often faced major issues (sequelae and/or death) for aspiration than anaphylaxis (27% and 5%, respectively). The respondents subject to legal litigations were rare (3% and 4%).

### 3.5. Difference in RSI Practice Between Countries

The complete results are available in the Appendix A. To avoid too much heterogeneity in the results, only countries with at least 10 respondents were analyzed, accounting for 66% of the respondents (327/494). All countries were in Europe except India and Turkey. Briefly, among the 15 countries, 9 had more than 70% of respondents using opioids in addition to short-acting neuromuscular blockers and hypnotics for RSI. Conversely, more than 70% of respondents did not consider opioids in France and Turkey. Practices concerning preoxygenation were homogeneous, with the use of a high flow of oxygen for most of the respondents, despite the use of alternative support varying. Indeed, non-invasive ventilation was unfrequently considered, in fewer than 50% of respondents, essentially in France, Austria, Sweden and Switzerland. A high-flow nasal cannula was rarely considered, with a maximal of 45% of respondents in Austria, and below 30% for the other countries.

## 4. Discussion

The current declarative survey explored practices concerning RSI in a wide population of anesthetics, of whom the majority were senior physicians. We observed that RSI definition was homogeneous among respondents, and interestingly most of them considered opioids as part of the pharmacological options. Moreover, RSI use seemed accepted for the classical indication at high risk of aspirations, notably during bowel obstruction, caesarean section or emergency surgery without respecting fasting. As expected, more controversial indications were subjected to heterogeneity, such as esophageal reflux or pain, suggesting the need for additional studies on these topics. Another result of importance is that almost a third of the respondents have already been exposed to serious consequences such as anaphylaxis or aspiration, despite rare situations of legal litigation. This reinforces the need for structured guidelines. Several results must be discussed in relation to the current guidelines and literature.

### 4.1. RSI Definition

The intent of RSI is to shorten the period between loss of consciousness and optimal conditions for intubation, allowing the absence of facemask ventilation, thus reducing the risk of aspiration. Thus, RSI definition may include all the processes that allow anesthesia induction with a minimal risk of aspiration. It comprises drugs choice, preoxygenation techniques and other aspects that may minimize the risk [11]. In a 2020 survey including almost 2000 respondents, the preferred drugs were propofol and succinylcholine in hemodynamically stable patients [9]. This historical association of a hypnotic with a short-acting neuromuscular blocker has been recently challenged in a multicenter randomized trial comparing the use of a neuromuscular blocker versus the opioid remifentanil [12]. In this trial, remifentanil was inferior to neuromuscular blockers (essentially succinylcholine) for first-attempt intubation. No difference was observed for hemodynamic instability or anaphylaxis between groups. Thus, the use of remifentanil as an alternative to neuromuscular blockers does not seem adequate to date. Nevertheless, because of its impact on the cardiovascular system, the use of an opioid may be questioned in addition to RSI. In our survey, most of the respondents declared frequently using opioids in addition to hypnotics and short-acting neuromuscular blockers. The addition of remifentanil to RSI has been previously described. Its use for cesarean sections during preeclampsia reduced cardiovascular responses to nociception, despite a transient neonatal respiratory depression [13]. In a randomized trial on elderly patients, the addition of remifentanil at doses inferior to 1 µg/kg/min resulted in a reduction in hypertension events, but with an increase in the incidence of hypotension in up to 24% of cases [14]. Taken together, these data suggest that the addition of opioids to RSI may reduce adverse cardiovascular responses to tracheal intubation, but with a higher risk of cardiovascular depressions. The benefit of its use must therefore be assessed according to the individual and clinical setting. In our survey, 69% of respondents declared regularly using an opioid during RSI. This result is similar to two previous surveys where 80 and 69% of respondents declared using opioids [8,9].

Interestingly, over 70% of respondents had encountered an aspiration situation in their professional practice, while fewer than 30% had experienced severe anaphylaxis. Both rates are high but may reflect the higher incidence of aspiration compared to anaphylaxis. Legal issues were nevertheless rare.

### 4.2. Preoxygenation

Concerning preoxygenation, most of the respondents followed the current guidelines from anesthesiology societies, with the use of a high flow of oxygen before induction (>10 L/min) [11,15,16]. Other techniques were non-invasive ventilation with pressure support with (17%) or without a positive end-expiratory pressure (11%), which has proven its efficacy in shortening the preoxygenation period and reducing the incidence of arterial desaturation after anesthesia induction, especially in hypoxemic patients [17,18]. A high-flow nasal cannula (HFNC) is also described for preoxygenation but the results from the current studies are heterogenous. If HFNC improved oxygenation in a before–after study in ICU patients, the results from a randomized trial in a population of hypoxemic patients did not reach similar results, but its use was associated with a reduction in adverse effects during intubation [19,20,21]. Its use may be relevant in particular populations, such as obese patients, where the functional residual capacity is reduced and the use of apneic oxygenation may be relevant, as suggested by some guidelines [22]. In our survey, 20% of respondents use HFNC for preoxygenation in the operating room, which is low in comparison with the ICU, where 84% of respondents declared using it before intubation in a 2019 French survey [23].

### 4.3. Cricoid Pressure

Cricoid pressure was first described by Sellick in 1961 to prevent the regurgitation of gastric content during RSI [6]. The efficacy of this maneuver is actually debated. In 2015, a systematic review did not observe benefits for the Sellick maneuver during RSI [24]. In a 2019 multicenter randomized double-blind study on 3472 patients, cricoid pressure did not reduce the risk of pulmonary aspiration [25]. In this trial, the majority of patients were at risk of a full stomach. Thus, 40% of our respondents use the Sellick maneuver during RSI, which is far lower than in a previous survey where 92% of respondents declared using it [8]. Despite the low evidence of this practice, and its less frequent use across decades, RSI is still suggested or even recommended in some guidelines, such as those from the universal management airways project, highlighting the hot debate around this topic. Nevertheless, in relation to the recent randomized trial with no reduction in aspiration risk, it appears that cricoid pressure is of no (or little) interest in most RSI situations. In particular situations of high risk, i.e., bowel occlusion, its use may be discussed because few data are available on this specific topic.

### 4.4. Suxamethonium and Rocuronium

Recent guidelines currently recommend the use of both suxamethonium or rocuronium for RSI [26]. Similar conditions for intubation were observed in a systematic review [27]. Over 99% of our respondents use a neuromuscular blocker for RSI induction but the choice between suxamethonium or rocuronium may depend on many factors such as the practitioner’s experience, cost and availability of sugammadex. To date, there are no data to argue for one or the other in the majority of clinical settings, with the exception of the specific contradictions of the two drugs (hyperkaliemia, severely burned patients, prolonged lying position or myopathy for suxamethonium, and severe renal failure for rocuronium).

### 4.5. RSI for Particular Situations

#### 4.5.1. Abdominal Emergencies

Acute abdominal situations are at risk of aspiration because of an increase in intragastric content, and its occurrence is well documented as being associated with mortality during anesthesia [4]. As expected, the majority of respondents declared placing a gastric tube and performing RSI in such situations. These results are consistent with a 2020 international survey [28]. Contrariwise, the results for appendicectomy are heterogeneous, with only 50% using RSI when patients are presenting no clinical sign of a “full” stomach. This result differs from a retrospective analysis on 250 cases in 2005 in a Canadian tertiary hospital, where 81% of cases were handled using RSI [29]. The question of RSI for appendicectomy is still debated. Indeed, intraoperative aspiration has been documented to be rare in patients with fasting over 8 h (0.1%), and the incidence of pulmonary complications was not influenced by the use of RSI [30]. Nevertheless, the absence of a robust literature and guidelines on this subject may explain the heterogeneity in practices.

#### 4.5.2. Cesarean Section

General anesthesia for cesarean section is associated with worse neonatal outcomes and vital risk for pregnant women [31]. Indeed, patients are at risk of difficult airway management, which, when combined with the reduced functional residual capacity, exposes to a high risk of hypoxemia, and then justifies RSI [32]. Hormonal modifications and the increase in intra-abdominal pressure are responsible for the incompetence of the lower esophageal sphincter and slowing of gastric emptying, all of which increase the risk of aspiration [33]. Surprisingly, only 80% of the respondents chose RSI for cesarean delivery and 66% of them use gastric acid buffering before induction, whereas both practices are suggested by the current guidelines [34].

#### 4.5.3. Gastroesophageal Reflux and Reduced Gastric Motility

Over 70% of our respondents use RSI in cases of symptomatic gastroesophageal reflux but nearly 30% still use standard induction, which is not in line with the current guidelines [35]. The aim of the survey was an inventory of the current practices in different countries; we did not ask the practitioners to justify their clinical strategies. Several hypotheses may explain this discrepancy with the guidelines. Firstly, some drugs may not have been available in some centers. Secondly, the past history of each practitioner may have modified their practice, despite the guidelines. Indeed, nearly 30% of respondents declared having already been involved in a grade 3 anaphylaxis due to a short-acting neuromuscular blocker, which may have modified their practice.

On the other hand, some patients may be at risk of aspiration because of persistent gastric content, even after several hours of fasting. It may be the case of unbalanced diabetes, obesity, recent or chronic administration of opioid analgesics or trauma [36]. In our survey, RSI was used in 45% of cases for bariatric surgery. This low rate may be supported by studies where the gastric content was not different in obese and non-obese patients [37], which supports the recent guidelines where airway management should not differ from non-obese patients, except in cases of reflux or gastric motility disturbance [38]. For example, the frequent association with diabetes mellitus may expose to a specific risk. Indeed, long-lasting and unbalanced diabetes may reduce gastric motility and emptying, even after several hours of fasting [39,40]. Recently, the use of certain medications in the management of diabetes, such as glucagon-like-peptide-1 (GLP-1) receptor agonists, has raised some concerns about gastric-emptying delay. The recent guidelines about the preoperative assessment of adults undergoing elective noncardiac surgery recommended patients treated with GLP-1 agonists should be considered at risk of aspiration despite a lack of gastrointestinal symptoms [41]. In our survey, 87% of respondents used standard induction in the absence of clinical signs of gastroparesis. Because of the potential side effects of RSI, this practice seems adequate, on condition of a rigorous evaluation and search for signs of gastroparesis. Gastric ultrasound may be of interest in this specific setting. Preoperative gastric ultrasound has been shown to be simple and efficient to identify gastric content before RSI [42]. Nevertheless, most of the respondents did not use this technique, mostly because of a lack of training.

Trauma may also delay gastric emptying and negatively affects fasting status [43]. In our survey, most of the practitioners used RSI for bimalleolar fracture (81%). Similarly, gastric ultrasound may be of interest in this specific population, but if this technique can describe gastric content, further studies are needed to evaluate its efficiency in the reduction in aspiration events [44].

#### 4.5.4. Survey Strengths and Limits

In this survey, we explored practices through several clinical vignettes. To our knowledge, previous surveys only explored factual declarative practices. In our opinion, our approach allows for a better analysis of RSI practices, particularly in non-consensual situations, providing a useful picture for the development of future guidelines. Moreover, respondents were essentially senior practitioners (79%), with 69% presenting with at least 10 years of professional experience in anesthesia or critical care. But there are also some limits that must be addressed. Firstly, despite a high number of responses and the different nationalities of responders, the declarative nature of the survey exposes itself to a high risk of bias and might not be representative of a global practice. Responses could be different in other hospitals or facilities and in non-members of the ESAIC (or other societies). Secondly, we did not separate the low-income countries from the high-income countries. This could impact the practice and change the choice of the anesthetist because of economic reasons. For example, many countries do not have access to sugammadex, rocuronium or gastric echography.

## 5. Conclusions

The definition of RSI seems homogeneous among respondents, with a majority of them including opioids as a usual drug, despite regional variations. The use of RSI seems to be firmly anchored in its historical use, i.e., in cases of bowel obstruction. Contrariwise, practices appeared to be heterogenous for several clinical situations where the literature and guidelines are rare or lacking. Complications like pulmonary aspiration seem surprisingly frequently declared in most of the countries in our survey. In conclusion, this survey highlights the need for practice guidelines and suggests maintaining efforts in exploring these areas of research.

## Figures and Tables

**Table 1 jcm-14-02177-t001:** Practices for rapid sequence induction according to respondents. Short-acting neuromuscular blockers comprising succhinylcholine and rocuronium. *: at least one physician with another physician, specialized nurse or resident.

Drugs	
Short-acting hypnotic + short-acting neuromuscular blocker + opioid	69% (342/494)
Short-acting hypnotic + short-acting neuromuscular blocker	49% (243/494)
Short-acting hypnotic + opioid	1% (8/494)
Short-acting hypnotic alone	1% (6/494)
Others	5% (24/494)
Preoxygenation	
High-flow oxygen (>10 L/min), without PEEP	87% (400/458)
High-flow nasal cannula	20% (93/458)
NIV with pressure support, with PEEP (≥3 cmH_2_O)	17% (79/458)
NIV with pressure support, without PEEP	12% (53/458)
Other practices	
Presence of at least two anesthesia providers *	75% (342/459)
Stylet inside the tracheal tube	65% (299/459)
Sellick’s maneuver	4% (191/459)

NIV: non-invasive ventilation; PEEP: positive end-expiratory pressure.

**Table 2 jcm-14-02177-t002:** Responses to different clinical vignettes where respondents were asking for general anesthesia management. PPI: proton pump inhibitor.

	Standard Induction	Rapid Sequence Induction	Gastric Suction Before Anesthesia	Gastric Buffering Before Anesthesia	Prokinetic Drugs Before Anesthesia	None of the Proposals
43 y/o laparoscopy appendectomy fasting > 12 h no vomiting	53% (236/447)	46% (204/447)	3% (12/447)	15% (69/447)	12% (55/447)	2% (7/447)
78 y/o acute bowel obstruction vomiting	0% (2/447)	97% (433/447)	83% (370/447)	34% (150/447)	18% (80/447)	0% (2/447)
24 y/o planned caesarean section	9% (38/447)	89% (397/447)	2% (8/447)	66% (295/447)	31% (137/447)	1% (5/447)
22 y/o bimalleolar fracture 2 h after eating	1% (4/447)	81% (364/447)	10% (44/447)	26% (115/447)	24% (107/447)	17% (76/447)
44 y/o osteosynthesis for tibial fracture Time between trauma and last meal 2 h—time between last meal and surgery 10 h	51% (227/447)	41% (182/447)	1% (6/447)	20% (91/447)	15% (65/447)	9% (40/447)
78 y/o prostate resection unbalanced diabetes mellitus—no sign of gastroparesis	87% (387/446)	9% (42/446)	2% (7/446)	18% (80/446)	12% (52/446)	3% (15/446)
62 y/o arterioembolization for cerebral hemorrhage (aneurysm) VAS pain 5/10—no nausea	68% (305/447)	30% (136/447)	1% (3/447)	19% (85/447)	10% (43/447)	2% (8/447)
56 y/o thyroidectomy untreated symptomatic gastroesophageal reflux	29% (131/447)	70% (314/447)	2% (10/447)	54% (242/447)	24% (107/447)	1% (4/447)
56 y/o thyroidectomy Asymptomatic gastroesophageal reflux under PPI	72% (321/447)	26% (115/447)	1% (3/447)	22% (99/447)	14% (64/447)	2% (7/447)
31 y/o bariatric surgery BMI 48 kg/m^2^—no gastroesophageal reflux	51% (227/447)	44% (198/447)	3% (15/447)	30% (136/447)	21% (92/447)	5% (22/447)

**Table 3 jcm-14-02177-t003:** Declared aspiration events and RSI-related anaphylaxis (due to rocuronium or succinylcholine), and the potential legal litigations associated with these events.

RSI side effects	
Anaphylaxis grade 3	29% (127/442)
Anaphylaxis grade 4 (cardiac arrest)	11% (47/443)
Major issues due to anaphylaxis (including mortality)	5% (11/208)
Aspiration	
Occurrence	76% (334/442)
Major issues due to aspiration (including mortality)	27% (100/372)
Legal litigations	
Related to anaphylaxis	4% (18/436)
Related to aspiration	3% (13/440)

## Data Availability

Data are available on reasonable request by contacting the corresponding author.

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
