# Peer review of "Practices of Rapid Sequence Induction for Prevention of Aspiration—An International Declarative Survey"

_jcm, 2025, doi:10.3390/jcm14072177_

Round 1

Reviewer 1 Report

Comments and Suggestions for Authors

This is a well-conducted international survey study examining current practices of Rapid Sequence Induction (RSI) across multiple countries. The study addresses an important clinical question given the lack of international guidelines and potential practice variations. The manuscript is generally well-written and provides valuable insights into real-world RSI practices.

Major Strengths:

  1. Large international sample size (491 respondents from 61 countries)
  2. Comprehensive assessment of RSI practices through detailed clinical vignettes
  3. Clear presentation of results with appropriate tables
  4. Good discussion of findings in context of current literature
  5. Practical implications highlighted for guideline development

Major Concerns:

  1. Response rate and potential selection bias not addressed - unclear how many practitioners received the survey
  2. Limited analysis of geographic/regional practice variations despite having data from 61 countries
  3. No power calculation or statistical analysis plan presented
  4. Incomplete reporting of some demographic data (e.g., exact distribution by country)

Specific Comments:

Methods:

  • Survey validation process could be better described
  • Timeframe (1 month) may be too short to capture seasonal variations in practice. Please explain this choice.
  • No description of how missing data was handled
  • Statistical analysis methods not described

Results:

  • Consider adding multivariate analysis to identify factors associated with RSI practice patterns
  • More detailed breakdown of respondent characteristics needed

Discussion:

  • Lines 261-263. Recent european preoperative evaluation guidelines have provided updated recommendations regarding preoperative assessment of patients at risk of aspiration. Please discuss Lamperti et al. EJA 2024.
  • More discussion needed on impact of resource availability on practice variations
  • Additional discussion of training/education implications warranted

Author Response

We thank the reviewer for her/his comments. We tried to take into account most of them and we hope it improved the manuscript. Please find below a point-by-point answer.

Major Concerns:

Response rate and potential selection bias not addressed - unclear how many practitioners received the survey :

We thank the reviewer for this relevant comment. We added these informations line 98-100. For that, we asked ESAIC and it took a few weeks, partially explaining the delay in resubmitting the manuscript. We apologize for that.

Limited analysis of geographic/regional practice variations despite having data from 61 countries

We effectively obtained data from a large number of countries. Nevertheless, most of countries were represented by few respondents. To avoid irrelevant analyses or comparisons, we focused on countries with at least 10 respondents. Moreover, we estimated that comparison of countries with unbalanced number of respondents will conduct bias in the statistical analysis. For these reasons, we presented data for 15 countries, accounting for 327 respondents. We added a paragraph, lines 168-180, and more detailed results in the supplementary file 2.

No power calculation or statistical analysis plan presented

As it was a declarative survey with no planned comparison, we did not calculate an a priori number of respondents. Similarly, the intent of the survey was not to compare groups but to depict practices. We think that comparisons will not be relevant as the numbers of respondents in each subgroup will be unbalanced and not representative.

Incomplete reporting of some demographic data (e.g., exact distribution by country): We added the information in the supplementary file 2

Specific Comments:

Methods:

Survey validation process could be better described

The survey was also validated and modified by two experts from the ESAIC. We added this point line 65.

Timeframe (1 month) may be too short to capture seasonal variations in practice. Please explain this choice :

The period for survey dissemination was planned by the ESAIC, and because of several concomitant surveys, a second dissemination was not planned, explaining the short timeframe. Nevertheless, we think that this characteristic may also be a strength as it may provide more homogeneity in the respondents (practices may vary over time).

No description of how missing data was handled

Missing data were not imputed because of the declarative nature of the survey. This point has been added line97

Statistical analysis methods not described

We added a paragraph, lines 92-98

Results:

Consider adding multivariate analysis to identify factors associated with RSI practice patterns

We did not plan a multivariate analysis because the objective was to depict practice. It seems to us that declarative surveys are not appropriate to identify risk factors of events, as it may induce bias in selection of respondents.

More detailed breakdown of respondent characteristics needed

The complete characteristics are provided in the supplementary file 1

Discussion:

Lines 261-263. Recent european preoperative evaluation guidelines have provided updated recommendations regarding preoperative assessment of patients at risk of aspiration. Please discuss Lamperti et al. EJA 2024 :

We thank the reviewer for this relevant reference, we added in ref 41. We added a sentence line 309-313 on this point.

More discussion needed on impact of resource availability on practice variations:

Because the survey was disseminated through the ESAIC, most of respondents were from European countries (on even from UE) with relatively similar resources. Respondents from middle- or low-income countries are few, and we think the analysis of this point would not be relevant, based on our results. It may be the objective of another survey, where one of the main difficulties would be the efficiency of dissemination.

Additional discussion of training/education implications warranted :

79% of respondents were senior practitioners, with 69% of the total presenting at least 10 years of practice in the field. This point were presented in the supplementary file 1.We added a short sentence in the strengths of the survey, lines 331-333

Reviewer 2 Report

Comments and Suggestions for Authors

1、 On the first page, it is recommended to use the format "Department-Hospital-City-Country" for affiliations to improve clarity and consistency.

2、 Line 32: Correct the word “Devasting” to the more appropriate term “Devastating.”

3、 Introduction: Current guidelines on RSI in critically ill patients are already available (DOI: 10.1097/CCM.0000000000006000), and similar studies have been published previously (DOI: 10.1097/EJA.0000000000001194; 10.1093/bja/aew017). It is suggested to briefly introduce these references in the introduction and highlight the novel aspects of this study to differentiate it from prior works.

4、 In Table captions, please provide explanations for the abbreviations (e.g., NIV, PEEP) to ensure accessibility for readers unfamiliar with these terms.

5、 Quantitative data throughout the manuscript is expressed as mean ± SD. It is recommended to perform a normality test first. If the data does not follow a normal distribution, presenting it as mean ± SD may not be appropriate. Alternatively, you could use both mean ± SD and median (25%, 75% IQR) to better convey key data distributions.

6、 The current analysis relies mainly on descriptive statistics. Deeper comparisons or correlation analyses are missing. For example, is there a significant difference in RSI practices between countries? Are there key variables (e.g., experience level, hospital type) that impact RSI practices? Consider adding multivariate analyses to enhance the depth of the study.

7、 In the first paragraph of the discussion section, consider presenting your conclusions in order of importance, from most to least critical. This would provide a more structured and impactful summary of your findings.

8、 The conclusion does not adequately summarize the key findings of the study. It is recommended to provide a more comprehensive synthesis of the results and explicitly state the significance of this research for clinical practice or future studies.

Author Response

We thank the reviewer for her/his comments. We tried to take into account most of them and we hope it improved the manuscript. Please find below a point-by-point answer

1、 On the first page, it is recommended to use the format "Department-Hospital-City-Country" for affiliations to improve clarity and consistency:

We corrected accordingly

2 Line 32: Correct the word “Devasting” to the more appropriate term “Devastating.”:

done

3 Introduction: Current guidelines on RSI in critically ill patients are already available (DOI: 10.1097/CCM.0000000000006000), and similar studies have been published previously (DOI: 10.1097/EJA.0000000000001194; 10.1093/bja/aew017). It is suggested to briefly introduce these references in the introduction and highlight the novel aspects of this study to differentiate it from prior works:

We added this point lines 41-44

4 In Table captions, please provide explanations for the abbreviations (e.g., NIV, PEEP) to ensure accessibility for readers unfamiliar with these terms:

abbreviations are now explained

5 Quantitative data throughout the manuscript is expressed as mean ± SD. It is recommended to perform a normality test first. If the data does not follow a normal distribution, presenting it as mean ± SD may not be appropriate. Alternatively, you could use both mean ± SD and median (25%, 75% IQR) to better convey key data distributions.

Thank you for this relevant comment. We modified the presentation of age and number of anesthesia per week, as these values were not normally distributed. The data are now presented as medians and interquartile ranges, and it is also described in the statistics section.

6 The current analysis relies mainly on descriptive statistics. Deeper comparisons or correlation analyses are missing. For example, is there a significant difference in RSI practices between countries? Are there key variables (e.g., experience level, hospital type) that impact RSI practices? Consider adding multivariate analyses to enhance the depth of the study.

We effectively obtained data from a large number of countries. Nevertheless, most of countries were represented by few respondents. To avoid irrelevant analyses or comparisons, we focused on countries with at least 10 respondents. Moreover, we estimated that comparison of countries with unbalanced number of respondents will conduct bias in the statistical analysis. For these reasons, we presented data for 15 countries, accounting for 327 respondents. We added a paragraph, lines 168-180, and more detailed results in the supplementary file 2.

The intent of the survey was not to compare groups but to depict practices. We think that comparisons will not be relevant as the numbers of respondents in each subgroup will be unbalanced and not representative.

We did not plan a multivariate analysis because the objective was to depict practice. It seems to us that declarative surveys are not appropriate to identify risk factors of events, as it may induce bias in selection of respondents.

7 In the first paragraph of the discussion section, consider presenting your conclusions in order of importance, from most to least critical. This would provide a more structured and impactful summary of your findings

We added a paragraph lines 184-192.

8 The conclusion does not adequately summarize the key findings of the study. It is recommended to provide a more comprehensive synthesis of the results and explicitly state the significance of this research for clinical practice or future studies: We modified the conclusion accordingly

Reviewer 3 Report

Comments and Suggestions for Authors

I read with great interest the manuscript by ben naoui et al. on the practices of Rapid Sequence Induction across multiple countries. The study is sound and well written. However, I have some comments to make:

- The methods used for statistic analysis should be reported in details in a separate subsection of the methods section.

- In general, the methods section is too brief: how was the survey conducted? Authors should provide many more details.

- The titles of Table 1, 2 and 3 should be outside the table itself.

- I do not think that the discussion should be divided into subparagraphs.

Author Response

We thank the reviewer for her/his comments. We tried to take into account most of them and we hope it improved the manuscript. Please find below a point-by-point answer.

I read with great interest the manuscript by ben naoui et al. on the practices of Rapid Sequence Induction across multiple countries. The study is sound and well written. However, I have some comments to make:

- The methods used for statistic analysis should be reported in details in a separate subsection of the methods section.

We added a specific section

- In general, the methods section is too brief: how was the survey conducted? Authors should provide many more details :

more explanation of the conduction of the survey are provided

- The titles of Table 1, 2 and 3 should be outside the table itself: corrected

Done

- I do not think that the discussion should be divided into subparagraphs.

Done

Round 2

Reviewer 1 Report

Comments and Suggestions for Authors

The authors successfully addressed my concerns. I have no more comments to make.

Author Response

We thank the reviewer for her/his contribution in reviewing our manuscript

Reviewer 2 Report

Comments and Suggestions for Authors

Thank you for your thorough responses and revisions. I have no further questions. One small suggestion: please consider further refining the table layout (especially Table 2) to improve clarity.

Author Response

Thank you for your comments and the time spent in reviewing our manuscript. We modified the table accordingly

Regards